# Neutrophil to Lymphocyte Ratio: An Emerging Marker of the Relationships between the Immune System and Diseases

**DOI:** 10.3390/ijms23073636

**Published:** 2022-03-26

**Authors:** Agata Buonacera, Benedetta Stancanelli, Michele Colaci, Lorenzo Malatino

**Affiliations:** 1Emergency Department, Policlinico S. Marco, 95121 Catania, Italy; agata.buonacera@gmail.com; 2Unit of Internal Medicine, Policlinico S. Marco, 95121 Catania, Italy; benedetta.stancanelli@virgilio.it; 3Academic Unit of Internal Medicine and Hypertension Centre, Department of Clinical and Experimental Medicine, University of Catania, Cannizzaro Hospital, 95126 Catania, Italy; michele.colaci@unict.it

**Keywords:** neutrophil-to-lymphocyte ratio (NLR), marker, prognosis, mortality, sepsis, intensive care unit (ICU), coronavirus disease-2019 (COVID-19)

## Abstract

Over the last 10 years, the evaluation of the neutrophil-to-lymphocyte ratio (NLR) as an emerging marker of diseases has become a compelling field of bio-medical research. Although a precise and unique cut-off value has not been yet found, its role as a flag of immune system homeostasis is well established. NLR has a well-known prognostic value and independently correlates with mortality in the general population and in several specific subsets of disease (sepsis, pneumonia, COVID-19, cancer, etc.). Moreover, NLR was recently considered as part of the decision-making processes concerning the admission/recovery of patients with COVID-19 pneumonia. This review aims to provide an overview of the main use of this biomarker, focusing on the pathophysiology and the molecular basis underlying its central role as a reliable mirror of inflammatory status and adaptive immunity.

## 1. Introduction

The neutrophil-to-lymphocyte ratio (NLR), calculated as a simple ratio between the neutrophil and lymphocyte counts measured in peripheral blood, is a biomarker which conjugates two faces of the immune system: the innate immune response, mainly due to neutrophils, and adaptive immunity, supported by lymphocytes [1]. Neutrophils are responsible for the first line of host immune response against invading pathogens, through different mechanisms, including chemotaxis, phagocytosis, release of reactive oxygen species (ROS), granular proteins and the production and liberation of cytokines [2]. Neutrophils also play an important regulatory role in adaptive immunity and are the main effector cells during the systemic inflammatory response (SIRS). As regulators of innate immunity, neutrophils recruit, activate and programme other immune cells, secreting an array of pro-inflammatory and immunomodulatory cytokines and chemokines capable of enhancing the recruitment and effector functions of other immune cells, such as dendritic cells (DCs), B cells, NK cells, CD4, CD8 and γδ T cells, as well as mesenchymal stem cells [3].

An isolated rise in neutrophil count, and, consequently, an elevated NLR, can be observed in several conditions: bacterial or fungal infection [4,5,6], acute stroke [7], myocardial infarction [8], atherosclerosis [9], severe trauma [10], cancer [11], post-surgery complications [12] and any condition characterized by tissue damage that activates SIRS. This is because the early hyperdynamic phase of infection is characterized by a proinflammatory state, mediated by neutrophils and other inflammatory cells [4]. SIRS is associated with the suppression of neutrophil apoptosis, which augments neutrophil-mediated killing as part of the innate response [4]. Thus, NLR is often characterized by an increase in neutrophils and a decline in lymphocytes.

Moreover, NLR could predict mortality in the general population [1]. NLR was significantly associated with higher overall mortality (HR 1.14, 95% CI 1.10–1.17, per quartile of NLR) and also with specific causes of mortality, including heart disease (HR 1.17, 95% CI 1.06–1.29, per quartile of NLR), chronic lower respiratory diseases (1.24, 1.04–1.47), influenza/pneumonia (1.26, 1.03–1.54) and kidney diseases (1.62, 1.21–2.17) [1]. On the other hand, no significant associations of NLR with mortality due to cancer, cerebrovascular disease, accidents, or diabetes mellitus were noted [1]. In addition, in the Rotterdam study, it was shown that NLR levels were independently and significantly associated with an increased risk of all-cause mortality (HR 1.64; 95% CI 1.44-1.86) [13].

The early increase (<6 h) in NLR following acute physiological stress could confer on NLR the role of marker of acute stress earlier than other laboratory parameters (e.g., white blood cell count, bacteremia, C-reactive protein, CRP).

Otherwise, lymphocytes, in their variety (B cells; T cells CD4-positive, CD4/CD8-negative or CD8-positive; natural killer T cells), are especially responsible for adaptive immunity, providing an antigen-specific response regulated by the major histocompatibility complex (MHC) class I. Lymphocyte activity is involved in host response to viruses, tumor cells, atopy and in SIRS too [14]. Lower NLR is usually associated with favorable prognostic factors in every field of application, mirroring a preserved immune balance.

If NLR is a proven independent prognostic factor of morbidity and mortality in several diseases, its normal cut-off value is still under debate. Forget et al. [15], in a large retrospective case–control study, observed that normal NLR values in an adult, non-geriatric population in good health may be between 0.78 and 3.53. In the Rotterdam study [16], it was observed that the mean NLR in the general population was 1.76, with a 2.5% limit at 0.83 and 97.5% limit at 3.92. Moreover, the mean NLR was significantly higher in males (mean of 1.88) than in females (mean of 1.68) and in elderly subjects >85 years of age (mean NLR of 2.13) compared with subjects aged 45–54 years (mean NLR 1.63) (*p*  < 0.001) [16].

Karakonstantis et al. [17] found several factors that could determine a “false” increase in NLR, such as age, exogenous steroid intake, endogenous sexual hormones, active hematological disorders, such as leukemia, cytotoxic or granulocyte colony stimulating factor (G-CSF) chemotherapies, and HIV. Major NLR confounders are listed in Table 1 below.

Homeostasis between these two aspects of the immune response is often frail: it could be affected by many either physiological or pathological factors. As examples, endogenous cortisol and catecholamines may be major drivers of NLR. High levels of cortisol are known to increase neutrophil counts, while simultaneously decreasing lymphocyte counts. Likewise, endogenous catecholamines (e.g., epinephrine) may cause leukocytosis and lymphopenia. Cytokines and other hormones are also likely to be involved. Confounders must, therefore, be taken into account in order to assign the correct clinical meaning to NLR alteration.

In this review, we will examine the role of NLR in the most important fields of its application in human diseases.

## 2. NLR and the Pathophysiology of Inflammatory Disease

### 2.1. Sespis

NLR has been known as a reliable marker for the diagnosis of bacteremia and sepsis [18]. A recent metanalysis [19] showed that a higher NLR was associated with poor prognosis in patients with sepsis (mean HR 1.75) and that NLR was higher in non-survivors than in survivors from sepsis (mean HR 1.18). Moreover, in a single-center prospective observational study of septic patients admitted to an intensive care unit (ICU), NLR values (9.53 ± 2.31) correlated with sepsis severity as calculated with the SOFA score (R = 0.65) and also with presepsin (R = 0.56), with a sensitivity of 47%, a specificity of 78% and an AUC of 0.631 (*p* < 0.05) [20]. The same study also showed that NLR was significantly higher in patients with septic shock (10.31 ± 2.32), suggesting the potential value of NLR in assessing sepsis severity, especially when its value is above 10 [20]. Gurol et al. [21] proposed a cut-off value series for NLR according with procalcitonin (PCT) values as a tool for bacteremia or sepsis diagnosis, as well as for decision making. Compared with CRP and white blood cell count (WBC), NLR showed a moderate sensitivity (57.8%) and high specificity (84%). Several diseases associated with elevated NLR values, such as trauma, surgery, pancreatitis and rheumatic disorders [17], often associated with sepsis, may play confounding roles.

In the Gurol et al. study [21], patients were grouped according to the following criteria: PCT levels < 0.05 ng/mL (healthy group, HG), PCT levels 0.05–0.5 ng/mL (local infection group, LIG), PCT levels 0.5–2 ng/mL (systemic infection group, SIG), PCT levels 2–10 ng/mL (sepsis group, SG) and PCT levels > 10 ng/mL (sepsis shock group, SSG). The NLR cut-off values for the HG, LIG, SIG, SG and SSG groups were <5, ≥5–<10, ≥10–<13, ≥13–<15 and ≥15, respectively [21]. The authors proposed a NLR cut-off value of 5 for diagnosis of infection or sepsis when exclusion criteria are used [21]. A recent observational study demonstrated that NLR and IL-6 appeared to be independent predictors of 28-day mortality in septic patients [22]. Moreover, Jang et al. conducted a clustering analysis showing that age, NLR and delta neutrophil index (DNI) were the most robust predictors for sepsis status in all subjects, as well as in cluster-specific groups [23]. These findings could contribute to suggest a strategy to screen sepsis patients without leukocytosis in the emergency care units. This is because, as we said above, NLR increase precedes WBC and CRP alterations, being the first sign of the activation of the immune system during sepsis. However, there are insufficient data about the role of NLR to identify the source of sepsis.

### 2.2. Pneumonia

Community-acquired pneumonia (CAP) is one of the most common causes of sepsis needing hospitalization. It is characterized by high rates of mortality and morbidity, especially in elderly patients. NLR has demonstrated a strong predictive value concerning short- and long-term mortality, need for ICU admission and re-hospitalization [24,25,26,27,28]. Therefore, NLR correlated with post-CAP mortality better than traditional pneumonia scores (Pneumonia Severity Index, PSI; and Confusion, Urea, Respiratory rate and Blood pressure, aged 65 and older, CURB-65), WBC and CRP [24]. Patients with NLR greater than 28.3 had the worst prognosis, so it could be hypothesized that they would require admission to a respiratory ICU, while those with NLR less than 11.12 could be safely discharged and managed through an outpatient follow-up [24]. Furthermore, 3-month likelihood of rehospitalization increases in parallel with NLR increase, with a rate >75% when NLR is >20 [24]. Other authors have observed that not only NLR value but also its incremental change can predict clinical instability and all-cause 30-day mortality [26]. Of note, NLR values on admission did not differ significantly between survivors and non-survivors [26], while those at 3–5 days differed and predicted more accurately 30-day mortality [27]. This suggests that admission NLR values could not be used as a mortality prediction tool in hospitalized patients for CAP [26] or ventilator-acquired pneumonia (VAP) [28], due to the better performance of serial measurements. Recently, in a post hoc analysis of a randomized clinical trial (RCT) investigating adjunctive dexamethasone treatment in CAP, patients with NLR ≥ 15.5 experienced benefit from dexamethasone treatment in terms of hospital stay shortening by 2 days [29].

Considering nosocomial pneumonia, which is an emerging problem, especially in ICU, in a retrospective observational study [30] conducted with patients with hospital-acquired pneumonia (HAP), low NLR values were significantly correlated with multidrug-resistant (MDR) *Pseudomonas aeruginosa* (PA) (MDR-PA HAP) as well as with worst prognosis. The authors [30] claimed that one possible reason for this phenomenon was that the lower level of NLR may have been the consequence of a less virulent MDR-PA strain. Hence, it could be speculated that a decline in NLR (associated with other conventional tools, such as hemocultures) may indicate the involvement of a MDR strain [30]. Further studies are needed, however, to confirm the prognostic role of NLR in patients with HAP and VAP.

#### COVID-19 Pneumonia

COVID-19 pneumonia is responsible for often severe respiratory failure throughout the ongoing pandemic. To date, it is well known that the onset of SARS-CoV-2 infection is characterized by lymphocyte reduction and that the rate of this reduction inversely correlates with disease severity [31]. In particular, T and NK cells, which are necessary for control of viral infection, were markedly decreased, while B cells were at the lower level of their normal range [32]. Not only the number but also the function of NK and CD8+ T cells are compromised in COVID-19 patients. One of the reasons seems to be increased expression of the inhibitory receptor NKG2A in COVID-19 patients [33]. Of note, NKG2A is a heterodimeric inhibitory receptor prominently expressed by cytotoxic lymphocytes, such as NK cells and CD8+ T cells [34].

Importantly, after therapy, the number of NK and CD8+ T cells in COVID-19 patients was restored, with reduced expression of NKG2A. These results suggest that the functional exhaustion of cytotoxic lymphocytes is associated with the acute phase of SARS-CoV-2 infection [33]. By contrast, the percentage of naive T helper cells (CD3+CD4+CD45RA+) increased and memory T helper cells (CD3+CD4+CD45RO+) decreased in severe cases, as compared with mild-to-moderate cases [32]. Zheng et al. [33] demonstrated that breakdown antiviral immunity in COVID-19 patients is mediated by a consistent reduction in CD107a+ NK, IFN-γ+ NK, IL-2+ NK and TNF-α+ NK cell numbers and the mean fluorescence intensity (MFI) of granzyme B+ NK cells. In line with these findings, COVID-19 patients also showed decreased percentages of CD107a+ CD8+, IFN-γ+CD8+ and IL-2+CD8+ T cells and MFI of granzyme B+CD8+ T cells compared with healthy controls [33]. No significant differences were noted in the levels of IgA, IgG and complement proteins C3 or C4 between mild versus severe groups, while IgM decreased slightly only in severe cases. Furthermore, critical patients had higher levels of circulating inflammatory cytokines (e.g., IL-2R, IL-6, IL-8, IL-10 and TNF) and infection-related biomarkers (e.g., procalcitonin, serum ferritin and CRP) than patients with mild-to-moderate COVID-19 disease [32]. Likewise, the IFN-γ defense mechanism was impaired, meaning the lack of inhibition of IFN-γ on accumulation of pathogenic neutrophils and neutrophil survival in infected lungs, leading, in turn, to an accumulation of neutrophils [34]. This contributed to raise NLR values in patients with COVID-19, mirroring a detrimental activation of the immune system that becomes itself part of the disease. Other authors [35] demonstrated that in vitro monocyte infection by SARS-CoV-2 triggers caspase-1 activation, IL-1β and stimulates activation of the nucleotide-binding and oligomerization domain (NOD)-like receptor protein 3 (NLRP3, also known as “inflammasome”), thereby contributing to cell death. Furthermore, inflammasome activation in serum and postmortem tissues of COVID-19 patients was later shown by Rodrigues et al. [35].

As to the derangement of blood cell count, Sambataro et al. [36] recently demonstrated that COVID-19 patients showed a significant reduction of WBC count at admission to the ward and a more reduced neutrophil count relative to lymphocyte count compared with non-COVID-19 CAP patients [36]. The clinical implication is that WBC count could provide a simple and rapid tool for an early COVID-19 diagnostic triage in CAP patients.

NLR is the most studied biomarker in COVID-19. Its predictive value for progression of severe disease [37] and mortality [38] was largely confirmed by several studies [39,40]. In a study carried out in an ICU [41], non-survivors had significantly lower lymphocyte values (0.99 × 10^3^/mL (±0.87) vs. 0.77 × 10^3^/mL (±0.4), *p* = 0.003) and higher dNLR (7.13 (±3.9) vs. 8.77 (±4.58), *p* = 0.002) and NLR (12.34 (±9.54) vs. 15.4 (±9.43), *p* = 0.001) values. Moreover, NLR and dNLR showed the best independent predictive values for invasive mechanical ventilation need and death [41]. Data from our group [42] showed that in-hospital mortality in COVID-19 patients is predicted by a NLR > 11.38 (AUC = 0.771, *p* = < 0.0001, sensitivity = 77.5%, specificity = 65.9%).

Two recent large meta-analyses showed that severe and non-survivor COVID-19 patients had higher on-admission NLR levels than non-severe and survivor patients [43,44]. Regardless of the different NLR cut-off values, the pooled mortality RR in patients with elevated vs. normal NLR levels was 2.74 (95% CI 0.98–7.66) [44]. NLR cut-off ≥ 6.5 has an area under the curve (AUC) for mortality of 0.90, while NLR cut-off ≥ 4.5 has an AUC for disease severity of 0.85 [43]. Thus, NLR evaluation could allow an earlier diagnosis of severe cases which may reduce the overall mortality of COVID-19. Moreover, in a large cohort study including 12,862 individuals with COVID-19 [45], NLR has been proposed as a compass to predict corticosteroid therapy efficacy. In particular, admission NLR values > 6.11 correlated with more severe disease and a reduction in 60-day mortality rate in patients who received corticosteroid therapy. On the contrary, admission NLR values ≤ 6.11 correlated with mild-to-moderate disease that did not benefit from corticosteroid therapy [45].

### 2.3. Cardiovascular Disease

NLR has been shown to predict cardiovascular events in several studies. Shah et al. [46] found that in the general healthy population NLR > 4.5 independently predicts coronary heart disease (CHD)-related death long term (14 year). Moreover, the same study showed that NLR permits the accurate reclassification of those in the intermediate risk category of the Framingham Risk Score (FRS) as having a lower or higher probability of cardiovascular mortality [46]. Accumulating evidence has shown that NLR has a predictive value for disease severity and mortality in the spectrum of acute coronary syndromes (ACS) regardless of revascularization procedure and that it could be powered with association of other classical risk factors [47,48,49,50,51]. Furthermore, NLR prognostic values were studied also in diabetic patients, showing that NLR independently predicted major adverse cardiac events (MACEs) in these patients [52]. Our group [53] showed that in a cohort of 324 elderly patients, a NLR > 3.68 significantly predicted atherosclerotic carotid plaque formation. By contrast, recent data collected by Mannarino et al. do not support the role of NLR as a predictor of the risk of atherosclerosis progression and atherosclerotic cardiovascular disease (ASCVD) events in subjects at moderate-to-high ASCVD risk, in primary prevention. However, the usefulness of NLR for patients at a different level of ASCVD risk, as recognized by the authors themselves, cannot be inferred from this study [54]. Elevated NLR on admission, for particular values > 2.97 (sensitivity of 92.6% and specificity of 52.5%, AUC = 0.714, *p* = 0.001) was an independent predictor of all-cause 3-month mortality for hospitalized hypertensive patients older than 80 years [55]. The reason for NLR elevation in cardiovascular diseases is strictly linked to the well-known role of inflammation and oxidative stress in the pathophysiology of atherosclerosis and endothelial dysfunction [56]. In particular, the activation of the NPL3 inflammasome and consequent impairment in homeostasis between IL-1 and its antagonists was emphasized [56].

### 2.4. Cancer

Inflammation has a pivotal role in the pathophysiology of most solid and hematopoietic malignancies [57]. Inflammation induces tumorigenesis as a chronic pathological response to chronic infection, to immune disorders and to aging in predisposed subjects. Tumor initiation switches on the so-called “cancer-elicited inflammation (CEI)”, through a pro-inflammatory cytokine and chemokine storm, determine, in turn, the recruitment of immune cells, induction of angiogenesis and shifting to the promoting phase [57]. Stimulation of tumor-associated macrophages (TAMs), to secrete IL-1β, and of tumor-associated neutrophils (TANs) causes metastatic progression and potentiates systemic neutrophilic inflammation. This supports the notion that it can be considered a reliable and cheap marker of ongoing cancer-related inflammation and a valid indicator of prognosis of solid tumors [58]. The majority of meta-analyses have explored the prognostic value of NLR in various solid tumors. Templeton et al. [58], in their metanalysis, demonstrated that NLR higher than 4 independently predicts a shorter overall survival with many tumors (HR 1.81, 95% CI = 1.67 to 1.97; *p* < 0.001). Furthermore, NLR predicts cancer-specific survival (CSS), progression-free survival (PFS) and disease-free survival (DFS) with HR 1.61, 1.63, and 2.27, respectively (all *p* < 0.001) [58].

NLR may be also used for the stratification of cancer, correlating with tumor size, tumor stage, metastatic potential and lymphatic invasion.

Patients with glioblastoma and NLR lower than 4.7 had significant higher progression-free survival. As for other types of brain tumors, the predictive value of baseline NLR for brain metastasis was examined in stage IV non-small cell lung cancer patients and it was found that NLR higher than 4.95 correlates with a higher number of brain metastases. High NLR values were also associated with the presence at diagnosis or at follow-up of brain metastases, particularly in a group with adenocarcinoma [59]. The potential prognostic role of NLR concerning colorectal cancer remains controversial. In a multicenter survey of patients with urothelial cancer treated with immunotherapy [60], NLR with cut-offs ≥3 and ≥5 predicted progressive disease and poorer progression-free survival. Overall survival was predicted when NLR cut-off was ≥5 (*p* = 0.03) [60].

Finally, a recent review emphasized the role of NLR as marker of tumor progression, thus focusing on the role of neutrophils as potential targets of therapeutic strategies in the treatment of hepatocarcinoma [61].

Figure 1 summarizes the most reliable NLR cut-off values in the main pathophysiological states.

### 2.5. Surgery

Preoperative NLR values emerged as independent predictors for post-operative complications, as well as peri-procedural and post-procedural mortality, independently of the type of surgery (cardiac or abdomen surgery) [13,62,63]. However, patients undergoing surgery often show multimorbidity or multiple sources of stress, so that the real significance of NLR in these conditions could be misinterpreted.

## 3. Conclusions

NLR is a cheap and easy-to-obtain biomarker that mirrors the balance between two aspects of the immune process: (1) acute and chronic inflammation and (2) adaptive immunity. Even if, still, no fixed cut-off values are available, changes in NLR over time are a sign of immune system derangement. However, NLR could be considered a robust prognostic marker of disease severity and predictor of mortality, bearing in mind the effect of confounders and taking into account the specific context of the disease, comorbidities and therapeutic strategy. Further case–control studies could also help with the definite identification of its range of normalcy, adjusted for age categories. Finally, tertiles of range values, based on disease severity, instead of cut-off values, could increase the performance of this promising biomarker.

## Figures and Tables

**Figure 1 ijms-23-03636-f001:**
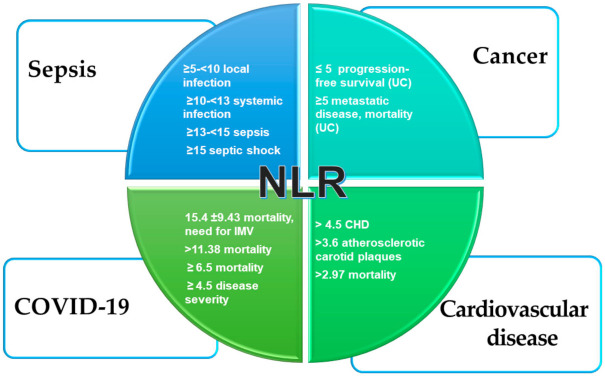
Reported NLR cut-off values and diseases. IMV, invasive mechanical ventilation; CHD, coronary heart disease; UC, urothelial cancer.

**Table 1 ijms-23-03636-t001:** Major condition that could determine a “false” increase in NLR values.

Major NLR Confounders
Age
Obesity
Exogenous steroid intake
Endogenous sexual hormone
Active hematological disorder
HIV
Acute myocardial infarction
Stroke
Pulmonary embolism
Type 2 diabetes
Cancer
Acute trauma
Emotional stress

## Data Availability

Not applicable.

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
