# Peer review of "Neutrophil to Lymphocyte Ratio: An Emerging Marker of the Relationships between the Immune System and Diseases"

_ijms, 2022, doi:10.3390/ijms23073636_

Round 1

Reviewer 1 Report

The authors in this concise review deal with an important and rather neglected up to now marker,the NLR ratio,in various disease states. The use of this marker is in its infancy ,but potentially usefull as an additional  factor that helps  clinical characterisation of patients. The review is concise ,not comprehensive ,but easy to study thoroughly and exploit the information it contains.

A minor point: The Significance of Neutrophil-to-Lymphocyte Ratio in Hepatocellular Carcinoma as a Prognostic Marker is omitted in 2.4 and have to be added Arvanitakis etal, Cancers 2021, 13, 2899. https://doi.org/10.3390/ cancers13122899

Author Response

We thank the Reviewer for the positive comments. We quoted the reference by Arvanitakis et al and added a sentence in the Discussion. The paper has been checked by a native English speaking Colleague.

Reviewer 2 Report

I read with great interest this work concerning a really important  prognostic 
marker of disease’s severity and a predictor of mortality. 

The authors well explained the relationship between NLR and Covid-19, the sepsis status, cardiovascular diseases and cancers. 

Well done

Author Response

we thank the Reviewer for the compliments. It has been greatly appreciated. The paper has been checked by a native English speaking Colleague.
